# Identifying Flow Pathways for Phosphorus Transport Using Observed Event Forensics and the CRAFT (Catchment Runoff Attenuation Flux Tool)

**Russell Adams** [1,2], **Paul Quinn** [2,*], **Nick Barber** [3] and **Sean Burke** [4]

1   Agri-Food and Biosciences Institute, 18a Newforge Lane, Malone Upper, Belfast BT9 5PX, UK; russell.adams@afbini.gov.uk
2   School of Engineering, Newcastle University, Newcastle Upon Tyne NE1 7RU, UK
3   Department of Geography, Durham University, Durham DH1 3LE, UK; nicholas.j.barber@durham.ac.uk
4   British Geological Survey, Nicker Hill, Keyworth NG12 5GG, UK; seanb@bgs.ac.uk
*   Correspondence: p.f.quinn@ncl.ac.uk

**Abstract:** Identifying key flow pathways is critical in order to understand the transport of Phosphorus (P) from agricultural headwater catchments. High frequency/resolution datasets from two such catchments in Northwest England enabled individual events to be examined to identify the flow (Q) and Total P (TP) and Total Reactive P (TRP) dynamics (forensics). Detailed analysis of multiple flow and water quality parameters is referred to here as the event forensics. Are there more flow pathways than just surface runoff (dominated by overland flow) and baseflow (mainly groundwater) contributing at the outlet of these catchments? If so, hydrograph separation alone will not be sufficient. This forensic analysis gives a classification of four storm event response types. Three classes are based on the balance of old and new water giving enrichment and dilution of TRP pattern in the subsurface flow. A fourth type was observed where a plume of nutrient is lost to the channel when there is no observed flow. Modelling is also essential when used in combination with the event forensics as this additional tool can identify distinct flow pathways in a robust form. A case study will apply the Catchment Runoff Attenuation Flux Tool (CRAFT) to two contrasting small headwater catchments in Northwest England, which formed part of the Demonstration Test Catchments (DTC) Programme. The model will use data collected during a series of events observed in the two catchments between the period 2012 and 2014. It has the ability to simulate fast near surface (that can represent flow in the upper soil horizons and field drains) and event subsurface soil flow, plus slower groundwater discharge. The model can capture P enrichment, dilution and the role that displacement of "old" P rich water has during events by mixing these flows. CRAFT captures the dominant flow and P fluxes as seen in the forensic analysis and can create outputs including smart export coefficients (based on flow pathways) that can be conveyed to policy makers to better underpin decision making.

**Keywords:** catchment hydrology; water quality modelling; phosphorus; flow pathways; near surface runoff

## 1. Introduction

Catchment modelling is a significant tool used for investigating water quality issues to provide evidence to inform policy makers and improve catchment management strategies [1]. Modelling can add to the knowledge base to better understand catchment dynamics, particularly the identification of flow pathways that can also affect water quality by transporting Phosphorus (P) and sediment from agricultural areas to the catchment outlet. This knowledge cannot always be obtained from observations and measurements (terrestrial or aquatic) alone in agricultural catchments [2–4]. Recently,

high resolution, in situ water quality data have become available [5–7], that can assist in both identifying the most appropriate model structure and the procedure of model calibration and validation itself.

Key hydrological and pollutant delivery processes that occur in small, temperate catchments include: critical source areas [2,8,9], variable source areas [10], overland flow [11], near surface flow/interflow [12], soil processes and leachate [11], subsurface stormflow [11], drain flow including land drainage [13], "shallow groundwater flux" / "return flow" [11], and lastly, deep groundwater flux [14]. These processes all impact on P losses through either mobilising high concentrations of P and/or transporting P to the catchment outlet (or other observation points). Seasonality, antecedent conditions, storm type (e.g., magnitude and duration) must all be considered when assessing the information provided by these observations [15–17]. This information must be coupled with a knowledge of farm types, landscape function and crop and nutrient management regimes where available [2]. Deciphering the information provided by the high-resolution nutrient data will be referred to here as event "forensics", however, this may be limited to a single location (catchment outlet) if distributed information on within-catchment P losses and/or sources is not available. Lastly, there exists a need to capture the dominant P loss activity in models at many sites and scales to avoid being too site specific.

It is therefore possible that analysing time series alone may not be sufficient to identify distinct flow pathways in the catchments as: (i) P is not necessarily conservative along each flow pathway and transformations may occur between soluble and insoluble forms as well as sorption/desorption within the soil; (ii) it was not possible to determine independently a priori Total P (TP) and Total Reactive P (TRP) concentrations unique to each flow pathway to perform End-Member Mixing Analysis (EMMA) [18], or separate the TP load time series into contributions from different flow pathways without using empirical methods that are subject to a high degree of uncertainty; iii) P may follow more than one flow pathway, e.g., surface runoff followed by infiltration through the soil and drain flow before it is measured at one location (catchment outlet). Jarvie et al. [19] attempted to use an EMMA approach using nutrient data (called *E-EMMA*) which could be more appropriate than the original EMMA for this study, if the required data can be determined.

Therefore, a key aim of this study is to observe P losses at the catchment scale (using data collected from sampling flow at the outlet over several years) and to rigorously test an event-scale model of flow and P transport that can be used to explore different hypotheses of flow pathways. To achieve this aim, this study will prioritise the underlying dominant processes by generating a set of outputs that can be transferred to other studies or act as a basis for scaling up and/or better-informed land management [20,21].

The rationale for this study is based on clear evidence that, globally, freshwater systems are vulnerable and under threat from high nutrient (P in this instance) loads [1]. European national-level water quality targets are set according to the statutes in the EU Water Framework Directive (WFD) 2000/60/EC [22] and have attempted to improve water quality with mixed results. For example, in England, in 2017, 68% of rivers and canals failed to reach "good" ecological status according to the Joint Nature Conservation Committee [23]. Diffuse pollution was often cited as the culprit, especially in the form of P and nitrogen losses from agricultural systems [5].

Recently, in-stream, high-frequency/resolution monitoring equipment has become more widely available for catchment investigations [6,7,17,24], which have been collecting extensive datasets. These datasets enable important research questions to be posed that this paper will address, such as: (i) Can flow pathways be identified from flow and concentration data alone? (ii) Is modelling required to discriminate between flow pathways during events? Longer term monitoring programmes (e.g., the Irish Agricultural Catchments Programme (ACP) [3,15,16] and the Demonstration Test Catchments (DTC) platform) [5,6] have been in operation for around a decade now and have captured the full range of environmental conditions in the catchment (e.g., periods of floods and droughts), however in the humid temperate catchments investigated in these programmes storm events have been found to

be the dominant time periods, where large transfers of sediments and nutrients to the watercourses occur [16,25,26]. Hence, the analysis of catchment behaviour during storms is now the key issue.

In terms of P (principally total phosphorus (TP) and reactive phosphorus (RP)) the key points from several case studies in agricultural catchments are that firstly the flow pathways for each form of P can be quite different, for dissolved and particulate forms of P [10,15,20,27,28]. The faster flow pathways, typically referred to as "surface runoff", capture the dynamics of P transfer during events but can be relatively short-lived. The first aim of this study therefore was to study the dynamics of P export (concentration and flux) during events using high-resolution monitoring data and see if any clear signatures can be interpreted that may relate to the different flow pathways in the catchments. Event forensics were then used to compare the responses of two different catchments, instrumented as part of the River Eden part of the DTC programme (EdenDTC) [6,25], during numerous storm events to see whether anthropogenic or other (e.g., geomorphological) factors are leading to different responses. If mitigation is required then these differences could influence the strategies taken in both catchments to reduce P loads and concentrations. In this case the sensitivity of flow pathways to current management practices and their potential mitigation options is required. For example, degraded soils may yield more surface runoff [8] and this could be addressed by increasing the soil infiltration capacity. However, increasing soil infiltration could in turn elevate TRP levels.

The second aim of the paper was to simulate the two catchments using a flow pathway based hydrological model. When selecting the model, it was found that one criticism of water quality models is that they contain too many calibrated parameters and possess equifinality [29]. Therefore, the spatially lumped rainfall runoff model—the Catchment Runoff Attenuation Flux Tool (CRAFT) was used to glean additional information on the flow pathways in the catchments [20,21]. The CRAFT is designed to be as parsimonious as possible. The model currently contains representations of three flow pathways and the ability to attenuate surface flows (to simulate the effect of either natural or added storage, such as sediment traps, in the catchment [21]). In Ireland, the SMART model [30] has been developed to simulate diffuse nutrient impacts on water quality also based on identifying subsurface flow paths (two extra flow paths were included—drain flow and deep groundwater flow), however, the nutrient component (nitrogen and P) from a pre-existing model (Integrated Catchment Model (INCA)) was used.

Two modelling studies have already been reported using data from Eden catchment DTC; the first applied a transfer–function data-based model (DBM) to the Newby Beck Catchment (NBC) only, in order to predict annual and storm transfers of discharge (Q) and TP loads [31]. This study addressed the uncertainties in the observed data, as well as the issue of equifinality and an overly complex model structure by using as simple a structure as possible. The DBM approach has its merits but is essentially site-specific, therefore transferring the results to other catchments would be difficult to do without re-running the analysis. The second modelling study was carried out using the Soil Water Assessment Tool (SWAT) to estimate daily flows and TP loads from the NBC catchment [32]. Despite relaxing the limits of acceptability (for "behavioural" parameter sets to be accepted), the results were poor and also affected by limitations in the model structure, especially the use of a daily timestep to model sub-daily high resolution data without appropriate processes being represented (e.g. entrainment of P by surface runoff and/or raindrop detachment). The authors concluded that the SWAT model was not appropriate to guide management of P in this catchment.

Therefore, a clear opportunity exists to use the CRAFT in order to overcome some of these pitfalls and provide policy makers with a tool for examining scenarios of land management and mitigation [21]. The model has an appropriate number of flow pathways, the flexibility to swap or remove pathways and works at the spatial and temporal resolution of the observed data, i.e., catchment scale and hourly data. The philosophy behind the development of the CRAFT was to mimic the physical processes of a catchment and its flow pathways using the minimum information requirement (MIR) approach [8,20,33], in this case by utilizing a set of linear storage–discharge relationships to represent outflows from four stores in the CRAFT (Figure 1). Three stores are associated with their flow pathway

to the outlet, with the fourth store being an attenuation store that is connected to the surface runoff pathway. The role of the surface store is also to partition effective rainfall into surface runoff and infiltration. Surface runoff includes "near" surface runoff that, in reality, may pass through the topsoil layers and exfiltrate in variably saturated areas in the riparian zone [8,11]. The term "flow" is preferred to "runoff" when referring to the CRAFT's flow pathways. Infiltration is then further divided into fast subsurface flow in the soil (which may include interflow) and slow groundwater flow (which may also consist of near constant "background" flow including effluent return flows from any point sources).

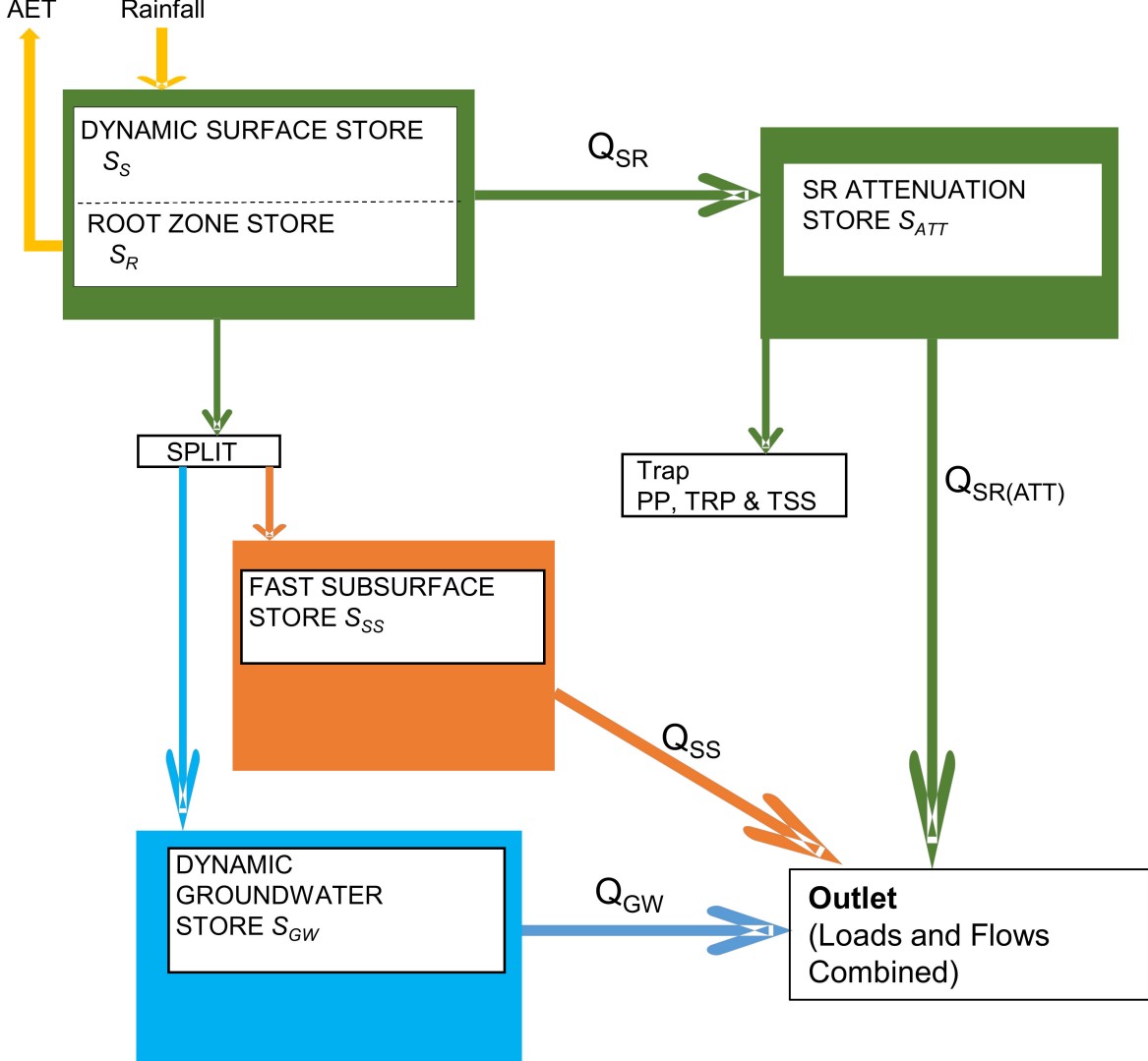

**Figure 1.** Schematic diagram of Catchment Runoff Attenuation Flux Tool (CRAFT) Model.

The Soluble Reactive P (SRP) and TRP concentrations in the three flow pathways are modelled by the CRAFT in order to predict the catchment SRP and TRP export. All RP in the slow groundwater and fast subsurface pathways is assumed to be soluble so will be referred to subsequently as SRP. Particulate P (PP) is also modelled in the surface runoff pathway only, this is typically seen as an enrichment of P from the soil to water [20]. The model structure will be described in more detail below (Section 2.3).

## 2. Materials and Methods

### 2.1. Description of Case Study

This case study used data collected from two instrumented sub-catchments investigated by the EdenDTC (Demonstration Test Catchments) project; these are located in the River Eden catchment in Northwest England [6,20,25,26], see Figure 2 below.

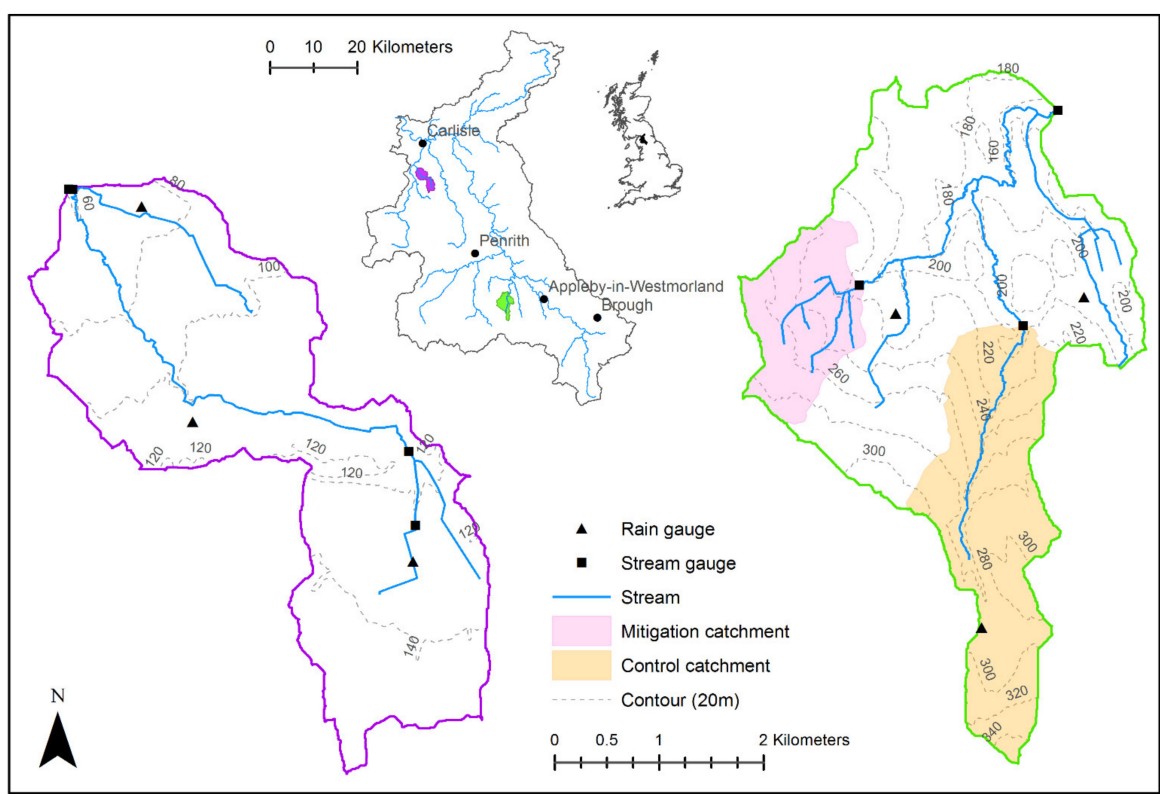

**Figure 2.** Map of the Eden catchment showing the two sub-catchments studied as part of the Eden Demonstration Test Catchments (EdenDTC) project, and the location of the monitoring sites for rainfall and flow in the sub-catchments. Contours at 20m intervals depict the topography (Newby Beck Catchment (NBC) green outline, Pow Beck Catchment (PBC) purple outline).

- Newby Beck. Firstly, the monitoring data collected at the Newby Beck Catchment (NBC) outlet at Morland were summarized by previous studies [6,20,25,26] and comprise 15-minute flow, electrical conductivity (EC), turbidity data and rainfall data and 30-minute TRP and TP concentrations. The NBC is very typical of the farm types and landscape of large proportions of the farmed area in the Eden catchment (Figure 2 bottom right).
- Pow Beck. Secondly, hydrological (Q and rainfall) and water quality data (TRP, TP, EC and turbidity) were also available from the Pow Beck Catchment (PBC) [26], at the same recording interval as the NBC data (Figure 2 bottom left).

Details on the land use, soils, hydrology, geology and geomorphology of both catchments can be found in Ockenden et al.'s study [26]. The key differences are that: (i) the PBC (mean slope 1°) is flatter than the NBC (mean slope 4°) and (ii) the PBC numerous gravel lenses [34] in the thick glacial deposits underlying the soil that act as temporary event stores that become connected to the stream network following heavy rainfall; (iii) the PBC has more intense agriculture including a diverse range of livestock farms and is thus a contrast to the NBC.

The high-resolution P data from both catchments were processed and any data gaps were identified, which accounted for around a quarter of the record in both catchments (Table 1). The period

between April 2012 and May 2014 was selected for analysis, as largely complete datasets were available for both NBC and PBC. Several runoff events were extracted for more detailed analysis, from more than 30 observed in the NBC and PBC meeting the criterion of having maximum event runoff $(q_p) \geq 0.3$ mm h$^{-1}$. There were periods of missing P data in either or both catchments so several events were omitted from the analysis. Loads over the entire period were adjusted for missing data by simply scaling up the total loads to account for the missing time period (e.g., if 20% of the data were missing then a scaling factor of 1.2 was applied).

**Table 1.** Observations at the NBC and PBC: April 2012 to May 2014, from monitoring data. Yields adjusted to provide an annual total. Total Unreactive Phosphorus (TUP) yield estimated by subtracting TRP yield from TP yield. Missing data were accounted for in yield calculations.

| Observation | NBC | PBC |
|---|---|---|
| Catchment Area (km$^2$) | 12.5 | 10.4 |
| Rainfall (mm) | 3057 | 2137 |
| Runoff (mm) | 1993 | 1355 |
| Runoff Coefficient (-) | 0.65 | 0.63 |
| TRP Yield (kg ha$^{-1}$ yr$^{-1}$) | 0.71 | 1.51 |
| TP Yield (kg ha$^{-1}$ yr$^{-1}$) | 1.86 | 2.45 |
| Estimated TUP Yield (kg ha$^{-1}$ yr$^{-1}$) | 1.15 | 0.94 |
| % Missing TP and TRP data | 23 | 27 |

Soluble P forms have not been measured in the EdenDTC subcatchments apart from a few discrete grab samples and some event samples collected using autosamplers. A study by Perks et al. [25] analysed the observed P data from the NBC (samples collected using the methods outlined above) and found that on average SRP concentrations were 86.2% of the TRP concentrations, the difference being accounted for by easily hydrolysable P species. Carrying out the same analysis on data from the PBC was also possible and the findings will be discussed further in the results below.

*2.2. Event Forensics*

Many studies rely on observed data to determine the process bases or the cause and effect of land management [2,7,16]. We refer to this as event forensics as the evidence is derived from piecing together many components of the data that can be identified through detailed analysis. The Irish ACP has clearly shown the power of this approach and the crucial role of flow pathways in P export [15,16]. The DTC programme has also pursued this approach and the case was made at the outset that observing processes at the event scale is key to better knowledge to underpin policy [5–7]. The simplest form of event forensics is the well-known technique of baseflow separation, which when automated can rapidly assess the percentage of total discharge that originated from baseflow vs. surface runoff (i.e. faster flow). However, as Singh and Stenger [35] pointed out, the technique is highly subjective and, without any additional information to identify what is "true" baseflow that originated from groundwater, may fail to accurately split runoff into two distinct components that are physically meaningful.

The high frequency/resolution data collected in the EdenDTC first allowed for detailed event forensics before any modelling of the data took place; this approach was used in the earlier studies, mainly in the NBC [6,20,25,26], and to a lesser extent in the PBC [26], but without any direct comparison of the two catchments being made. Having high temporal resolution TP and TRP concentrations and flows allowed for loads to be calculated during events. In addition in this study the load or concentration of Total Unreactive P (TUP) was calculated by subtraction of TRP from TP (load or concentrations). Here, it was assumed that TUP consists mostly of particulate P (PP) with some soluble, unreactive P consisting mainly of organic P forms.

Electrical conductivity (EC) data were available from the outlet of both catchments and can be used to identify periods of dilution due to fast flow pathways transmitting "new" event water into the stream network, based on the assumption that "new" event water has a lower EC value than

"old" water residing in the deeper soil layers [36]. Data collected via precipitation samples from the UK's Environmental Change Network (ECN) [37] were available from Moor House (less than 100 km from the Eden catchment) from the period 1992 to 2016 and indicated that the EC of precipitation was usually between 10 and 30 μS cm$^{-1}$. Therefore, it would be expected that during events the EC values measured in streamflow at the catchment outlets would be significantly less than the values measured for most of the time during inter-event periods (i.e., "baseflow") [36]. The EMMA assumptions [18] may be valid here if the bulk of the anions and cations being measured by EC are: (i) conservative along the flow pathways in the catchment, (ii) and do not change during the course of the event. However, such an analysis would only provide the proportion of "new" event water in the total runoff during the events without identifying pathways explicitly. The role of displaced "old" water also needs to be reflected in fast subsurface pathway and this can be studied in the recession component of the storm.

In order to compare the two catchments, statistical analysis was undertaken relating to event characteristics and these are shown below in Table 2 in the results. It was possible to classify events into different types based on the concentration and load dynamics, following on from the methods used in earlier studies [25,26,28], that mostly focused only on the NBC and identified the hysteresis relationships between both TP and TRP with Q, for a series of 55 events between the period May 2012 and February 2013. The Ockenden et al. study [26] investigated both catchments but did not compare them directly (different time periods were analysed). They used an event classification system (based on [28]) to distinguish three event types in both catchments. This complexity may suggest the need for a model of the catchments that can simulate multiple flow pathways, hence CRAFT was also used to determine whether the event dynamics were similar or different in the two catchments and this will be described in the next section. Here, we advance the classification of storm types based on TP, TRP and EC dynamics.

## 2.3. Modelling Using the CRAFT

The CRAFT was already calibrated for the entire NBC (over one year commencing from 1st October 2011) [20] in a modelling exercise where several simulations were carried out to test hypotheses of different conceptual models, which differed primarily in whether any attenuation was included in the model. The baseline scenario chosen for use in this study is the "lagged", in which the modelled hydrographs have added attenuation representing the natural storage in the catchment during runoff events. Validation was carried out using a period of 4 months between the period November 2012 and February 2013 [20], but for this study the model was also run for the period October 2013 to March 2014 without recalibration as a further validation test.

The calibration of the CRAFT to simulate runoff and P in the PBC was carried out using the "Expert" method used in the NBC modelling and model parameter values from the NBC were used as the starting estimates of the PBC values, although due to the different observed data periods the model was calibrated over one year of hourly data commencing on 1st April 2012. It was assumed *a priori* that the same model structure would be appropriate for the PBC, given the climatic and geomorphological similarities discussed above. The resultant simulation will be referred to subsequently as the PBC baseline simulation. The model was also validated over the period October 2013 to March 2014 without adjusting any parameter values.

Phosphorus Modelling

The P concentrations (per timestep) are extracted from each of the three pathways. To provide summary totals at an event or longer timescale, the CRAFT outputs can be summed to give a runoff depth and a nutrient load or yield (i.e. load/unit area) over the required period. The SRP loads from the fast subsurface and slow groundwater flow pathways are added to the loads of PP and TRP from the surface runoff pathway to compute the total load of TRP and TP (refer to Figure 1). The model does not simulate soluble forms of unreactive P in the surface runoff pathway so it assumes that PP ≈ TUP. These loads can be compared with the observed loads and a load error calculated

(predicted–observed)/observed. Issues of equifinality could arise if: (i) the loads and/or concentrations of SRP in the two subsurface flow pathways cannot clearly be identified from the observations (leading to different model parameter sets giving similar results in terms of concentrations at the outlet) and also (ii) if the split between the flow components along these pathways is not clearly identifiable from the runoff hydrographs. An earlier study [26] reported periods of time in both catchments when high concentrations of P were observed that were not associated with high flows (i.e., events), these were classed as "Type 3" events in Haygarth et al.'s analysis [28] and it is not possible to simulate these using the current CRAFT structure, if no flow pathway is being mobilised and recorded at the flow gauge. These enriched plume events in very low flows may be more associated with incidental losses or Critical Source Area (CSA) activity in very small storms.

A set of "smart" export coefficients (one for each pathway) which provide the modelled yields per flow pathway across the entire catchment, were also determined from the model output. These coefficients have an advantage over traditional export coefficients [12,38] that are purely quantifying the loss of nutrient per unit area without accounting for hydrological information (flow pathways) in a catchment as they are specific to the three pathways. Their values were calculated separately for the calibration period in 2012–2013 and the validation period in 2013–2014 which were both periods of high runoff and multiple events, and also for individual events.

In general, "expert" model calibration, as used in the NBC [20], aims to achieve a best fit to observed flows and P concentrations. Phosphorus concentrations during low-flow periods can be used as a guide to the SRP concentration in the slow groundwater pathway (providing most of the P observed originated from this source rather than point sources in the catchment). This procedure was also used when modelling small catchments with the TOPCAT-NP model [33]. It then gives the user of the models the option to calibrate the SRP concentration and the storage coefficient ($K_{SS}$) in the fast subsurface flow pathway in order to match the event P dynamics, which is useful if the implications of nutrient management policy is to be tested.

## 3. Results

### 3.1. Observed Data Analysis

Observed totals (flow, rainfall and nutrient loads) are shown in Table 1 for the time period of interest in this study which extended from the period April 2012 to May 2014 (see below for modelling results). For data analysis purposes (and modelling) the 15- or 30-minute interval observed data were averaged into an hourly timestep (or accumulated in the case of the 15-minute rainfall totals. The total runoff depth was calculated for the entire period and this enabled the runoff coefficients to be calculated (0.65 NBC vs. 0.63 PBC). Runoff coefficients of over 60% (Table 1) were relatively high although not unusual for wetter British catchments. The years 2012 and 2013 had an observed annual runoff rate of 60% and 62% resp. The clay rich soil type and boulder clay deposits are prone to waterlogging. The next part of the analysis relates to the high-resolution data set.

#### 3.1.1. Nutrient Data

The analysis of the ratio of SRP:TRP in the PBC was also calculated from SRP measurements made from autosampler and grab samples collected between the period April and September 2012 and found to be 0.94 (compared to 0.86 in the NBC [25]).

### 3.1.2. EC Data

The monitoring data revealed that baseflow periods had relatively high EC values ($>500$ $\mu$S cm$^{-1}$), this can be seen by the pre-event EC values in Figure 3 below. Event periods had much lower EC values due to dilution, suggesting the streamflow contained a high percentage of "new" event water. Data from groundwater samples collected in the NBC and PBC are summarised below in Table 2. These values indicated that in the NBC, EC was 554 $\mu$S cm$^{-1}$ (from one borehole only) and that in the PBC the EC varied between 380 and 1790 (mean = 884) $\mu$S cm$^{-1}$. These samples were collected on a single date only in August 2011 in both catchments. Therefore, the streamflow EC values during baseflow periods were much closer to their respective groundwater end members than the precipitation end member (10–30 $\mu$S cm$^{-1}$) in both catchments. In the PBC the difference was up to several hundred $\mu$S cm$^{-1}$.

**Table 2.** Details of the events observed the NBC and PBC between the period 2012 and 2014: Range of values indicated by parentheses followed by mean value of all event observations.

| Statistic | NBC | PBC |
|---|---|---|
| Event TP Maximum Concentration (mg P L$^{-1}$) | (0.17–1) 0.62 | (0.37–1.61) 0.81 |
| Event TUP Maximum Concentration (mg P L$^{-1}$) | (0.1–0.92) 0.49 | (0.11–0.84) 0.39 |
| Event TRP Maximum Concentration (mg P L$^{-1}$) | (0.08–0.38) 0.16 | (0.25–0.84) 0.44 |
| Mean Antecedent TRP (mg P L$^{-1}$) | 0.05 | 0.16 |
| Mean Increase in TRP (mg P L$^{-1}$) | 0.11 | 0.28 |
| Event $Q_p$ (mm h$^{-1}$) | (0.33–2.34) 1.08 | (0.34–3.03) 0.85 |
| Total Number of Events | 43 | 38 |
| Event Runoff (mm) | (2.6–37.4) 12.7 | (4.1–47.8) 12.6 |

### 3.1.3. Summary of Events

The event forensic analysis identified more than 30 events between April 2012 and May 2014; of these, 17 events were common to both catchments and are denoted by "NP<Event number>". The Supplementary Material contains plots of all 17 common events showing observed flow, TP and TRP concentrations (Figures S1 and S2). The event summary statistics are shown in Table 2 to provide information on the nutrient and runoff characteristics of the events. Runoff coefficients for individual events were highly variable in both catchments. Some important classifications of event types were made from the observations (time series plots of Q and P concentrations). Note that in the following plots, specific discharge (*q*) (i.e., runoff in mm/timestep) is used rather than Q, in order to compare the two catchments of different areas. Broadly, the events fell into the following categories based on their TP, TRP, EC and Q dynamics:

- These are classified as "enrichment" (E-type) events, where TP, TUP and TRP concentrations all increased with observed Q increasing up to a peak discharge value ($Q_p$) and correspond to the "type 2" events of Haygarth et al. [28]. EC falls across the rising limb and recovers during recession. Examples are shown in Figure 3 from the PBC; event NP2 (28th June 2012) and event NP6 (24th September 2012) (TUP is not shown for clarity but its pattern during these events closely mirrored that of TP). The responses suggest that "old" nutrient rich soil water is being displaced across the whole event. The displaced old water is a significant component of the overall TP level even during the peak of the event. The constant falling of EC was always observed as new water entered the channel. Clearly, old water must have been displaced first. More E-Type events were seen in PBC suggesting a larger nutrient P pool in the upper soil layers. Some mixing of old and new water probably occurred, but the pool of old water did not dilute during the event. The first part of NP2 in the NBC also exhibited E-Type behaviour (Figure 3 first row, top left)).

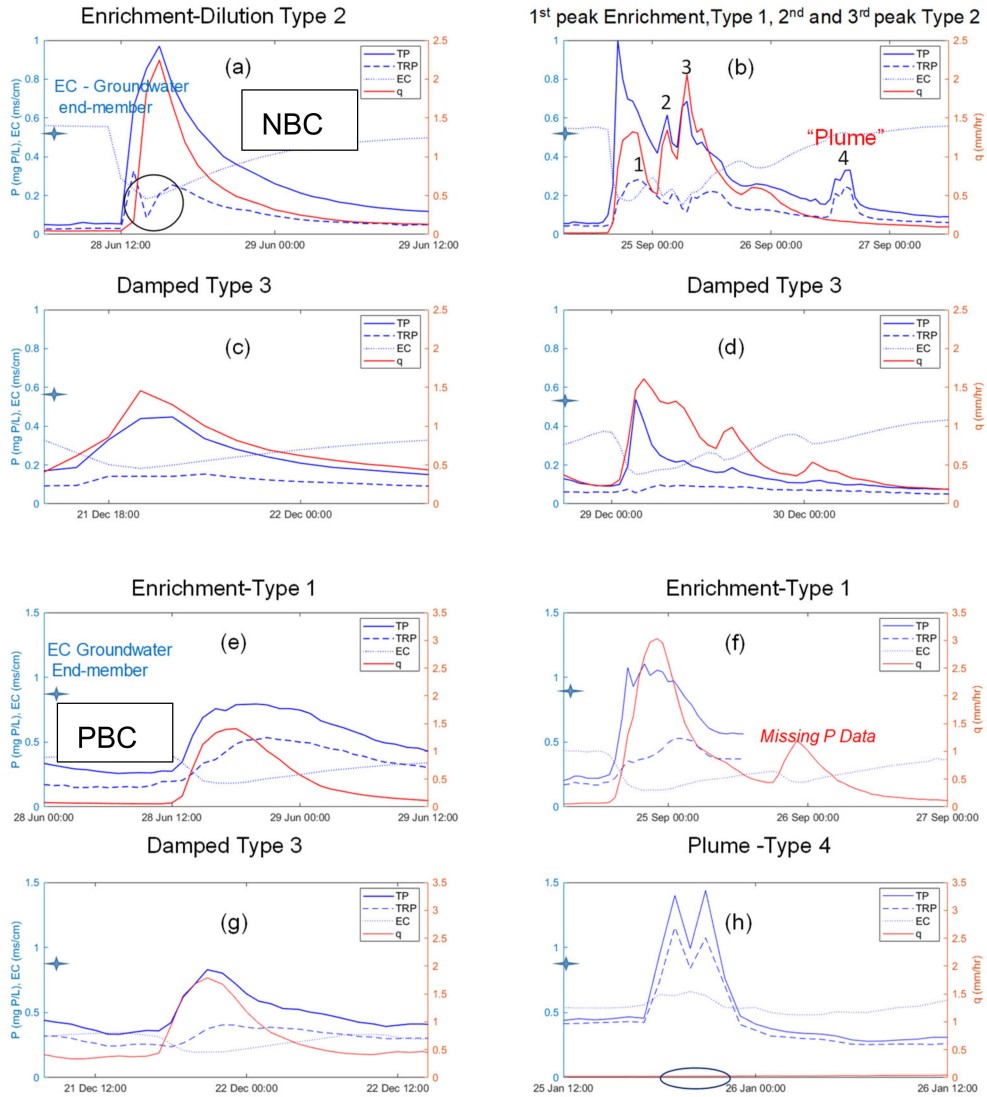

**Figure 3.** Time series plots of observed specific discharge (*q*), Electrical Conductivity (EC) (note that the EC units are mS cm$^{-1}$) and Total Phosphorus (TP) and Total Reactive Phosphorus (TRP) concentrations during events, indicating different event types. The first two rows (panes **a**–**d**) show four examples from the NBC; the last two rows (panes **e**–**h**) show four examples from the PBC. Specific discharge *q* is shown on the right-hand y axis.

- Enrichment–Dilution (ED-type) events, from the NBC example, are highlighted in event NP2 with the black circle in Figure 3 (pane (a)). Here, the TRP did not continue to rise and clearly dilution of Soluble Reactive Phosphorus (SRP) was occurring in the soil water pool. The falling EC trace was the same as found in E-Type events, but here a threshold was crossed where newer event water with lower TRP entered the channel via the fast subsurface pathway. This dilution also lowered the peak event TP concentration. The point at which the ED threshold was met is thus pivotal to the final response in the channel. This threshold for switching from E to ED Type was dependent on both the inter-storm duration and the magnitude of the storm events. A second example is shown in Figure 3 (pane (b)) in the NBC, initially there was an E-Type event, followed by two ED-Type events the following day. This demonstrates evidence that TRP was diluted during the second and third peak of the event, probably due to depletion of the readily available TRP pools after the first runoff peak had passed through the system. Remembering that a typical rainfall EC

value is of the order of tens of μS cm$^{-1}$, the decrease in EC to circa 150 μS cm$^{-1}$ probably indicated some in-stream dilution by "new" water.

- Damped (D-type) events. These are classified as "damped" type events where the TRP concentrations failed to increase in sync with either Q or TP concentration but stayed relatively low or constant throughout the event close to their pre-event levels. Two examples are shown in Figure 3 from the NBC from 21st December 2013 and 28th December 2013 (panes (c) and (d)), and one from the PBC on 21st December 2013 (pane (g)). In the NBC, TRP concentrations only rose by circa 0.05 mg P L$^{-1}$. In the PBC the TRP concentrations during these events failed to exceed 0.4mg P L$^{-1}$. These events were probably dominated by near surface runoff, suggesting either surface conditions reduced infiltration or that the upper portion of the soil water pool was depleted in SRP. Both events showed the same pattern of EC falling then recovering seen in the first two types.

- A "plume" event in the PBC was observed (Figure 3: pane (h)). Here, the nutrient plume occurred at very low rainfall and there was no detectable increase in flow. Thus, the forensic analysis and any model would struggle to simulate this event. The EC signal was most important here, it was the only occurrence found of EC rising during the event, i.e., no dilution. Clearly, the TP signal was dominated by this small but significant nutrient release that could have a high impact on eutrophication risk and ecology. This scenario is equivalent to the "type 3" event of Haygarth et al. [28]. These events seem to be rare especially in the NBC. Part 4 of the NP6 event in the NBC (Figure 3: pane (b)) could also be labelled as a plume event as flow was decreasing during the falling limb of the hydrograph when a "spike" of TRP was detected, but this was unusual.

In Figure 4, we see the flow pathways, firstly during the peak of an event (part A) and secondly during the recession (part B), these transport P during the runoff events discussed above. We can assume that only groundwater and a small amount of the soil water drains to the outlet in the inter-event periods. Here, we suggest that the enrichment in TP is driven primarily by the surface water flow pathway in all event types (except for "Plume" events). Figure 4 also shows the pivotal role the vertical recharge plays in controlling TRP concentration. The TRP flux pattern is caused by dynamic mixing of the old and new water during differing parts of the events. The observed data shows initially some displacement of "old" water ("E" and "ED" Types) but in the "D-Type" events the temporary influx of new water dominates the signal whilst it is still raining producing a damping effect. After the rainfall stops, the mixed "old" and "new" water now dominates. Any dilution in the EC signal is lost and EC then rises as the older water starts to dominate again as the recharge ends. This means that the "D-Type" signal pattern is only an artefact of the mixing ratio as seen in the "E" and "ED" Type events. All event types should have much higher peak TRP concentrations (as in the "E" Type examples) but for this dilution factor. This might also suggest that the "Plume" Type event on 25th January 2013 in the PBC (labelled "*Plume*" on Figure 3: pane (h)) fitted an example where surface runoff did not increase and no dilution occurred, but there was either enough recharge to displace the old P-rich source or a source of P related to land use practices (e.g., a leaking slurry tank). Thus, the dominance of the surface runoff pathway is clear, and more so in the NBC than the PBC. The damped and dilution signals are, in fact, the same process varying over time and controlled by either the greater soil depths in PBC or by the rainfall intensity–duration patterns. The tail end of the recession still includes the higher TRP concentrations, but there may be a small residual flow component coming from the faster surface pathway with some flow through the upper soil layers.

Both catchments exhibited variable responses in terms of the observed lag time between maximum P concentrations and $Q_p$ during events. Figure 5 shows a plot of the range of time lags observed from all events between firstly TP and $Q_p$ and secondly TRP and $Q_p$ and both positive (nutrient peaking after flow) and negative (nutrient peaking before the flow) lags during these events were observed. A negative time lag infers clockwise hysteresis where the concentration is higher on the rising limb than on the falling limb and a positive time lag an anti-clockwise hysteresis pattern (i.e., the converse). Sketches in Figure 5 depict these patterns. These responses tie in with flow pathways in the NBC and

PBC differing, i.e., NBC has less P export overall but is a very fast responding catchment. PBC has more P export but its deeper soils mean that P export is sustained for longer and its response is slower. Other phenomena picked up in hysteresis patterns will be dominated by the complex dilution ratio seen during the events as controlled by the thresholds of enrichment and dilution.

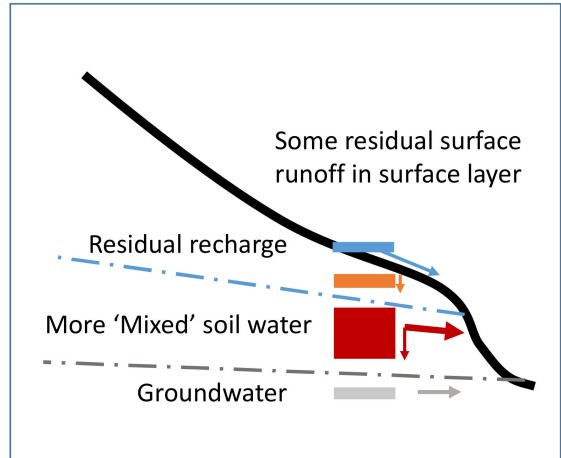

**Figure 4.** Conceptual model of Phosphorus (P) transfer by different flow pathways. Flow pathways and subsurface stores shown. Blue dashed line depicts the shallow water table (in soil), grey dashed line the groundwater table.

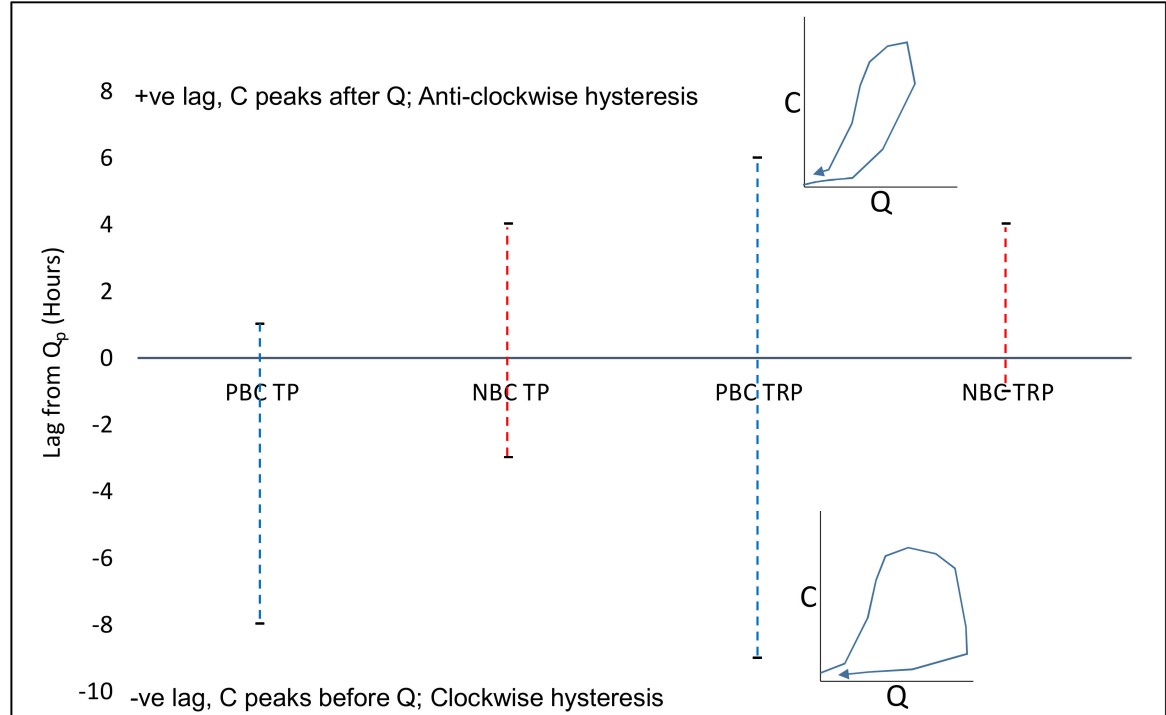

**Figure 5.** Plot showing the range of lag times for TP and TRP in NBC and PBC for all events. The lag times were calculated from the observed data from when the peak concentration (of TP and TRP) occurred relative to the time when $Q_p$ occurred. Sketches of Q-C relationships during hysteresis loops are also shown for clockwise and anti-clockwise hysteresis.

### 3.2. Modelling Analysis

Time series plots of observed and modelled runoff and rainfall for the periods April 1st–September 30th 2012, and 1st December 2013–31st January 2014, from the NBC and PBC are shown in Figure 6. The first time period was part of the model calibration period for both catchments whilst the second was part of an independent validation period.

The CRAFT was calibrated to a set of baseline conditions in both catchments and the NBC parameter values were not adjusted here from the values in the "Lagged" scenario in the previous modelling study [20]. Model parameters obtained through "expert" calibration were similar for both catchments, interestingly the value of the attenuation parameter $K_{LAG}$ obtained by calibration in the PBC was higher than the NBC value (0.8 h$^{-1}$ vs 0.75 h$^{-1}$) indicating that there may be slightly more natural attenuation in this sub-catchment. This could also be because the PBC is flatter than the NBC and has gravel lenses that act as temporary event stores contributing return flow after $Q_p$. The PBC thus exhibits a longer lag time between the runoff peak occurring in the hillslope and in the outlet hydrograph, which may be related to the superficial geology and slope [26,34].

The accuracy for the model for flow, TP and TRP fluxes are summarised in Table 3. The range of goodness of fit statistics shows that the model fits the flow very well, but it is much weaker for the concentrations of P species (TP and TRP).

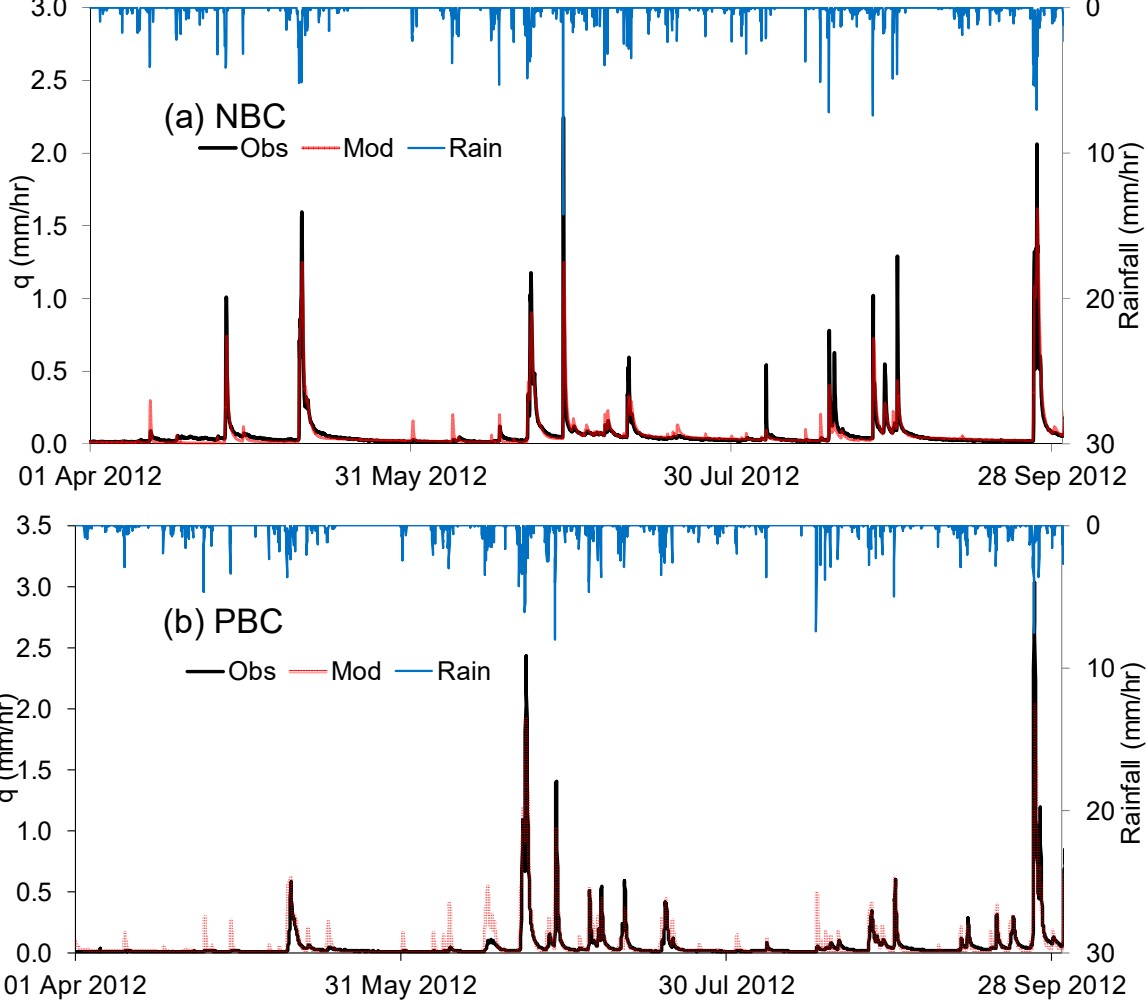

**Figure 6.** *Cont.*

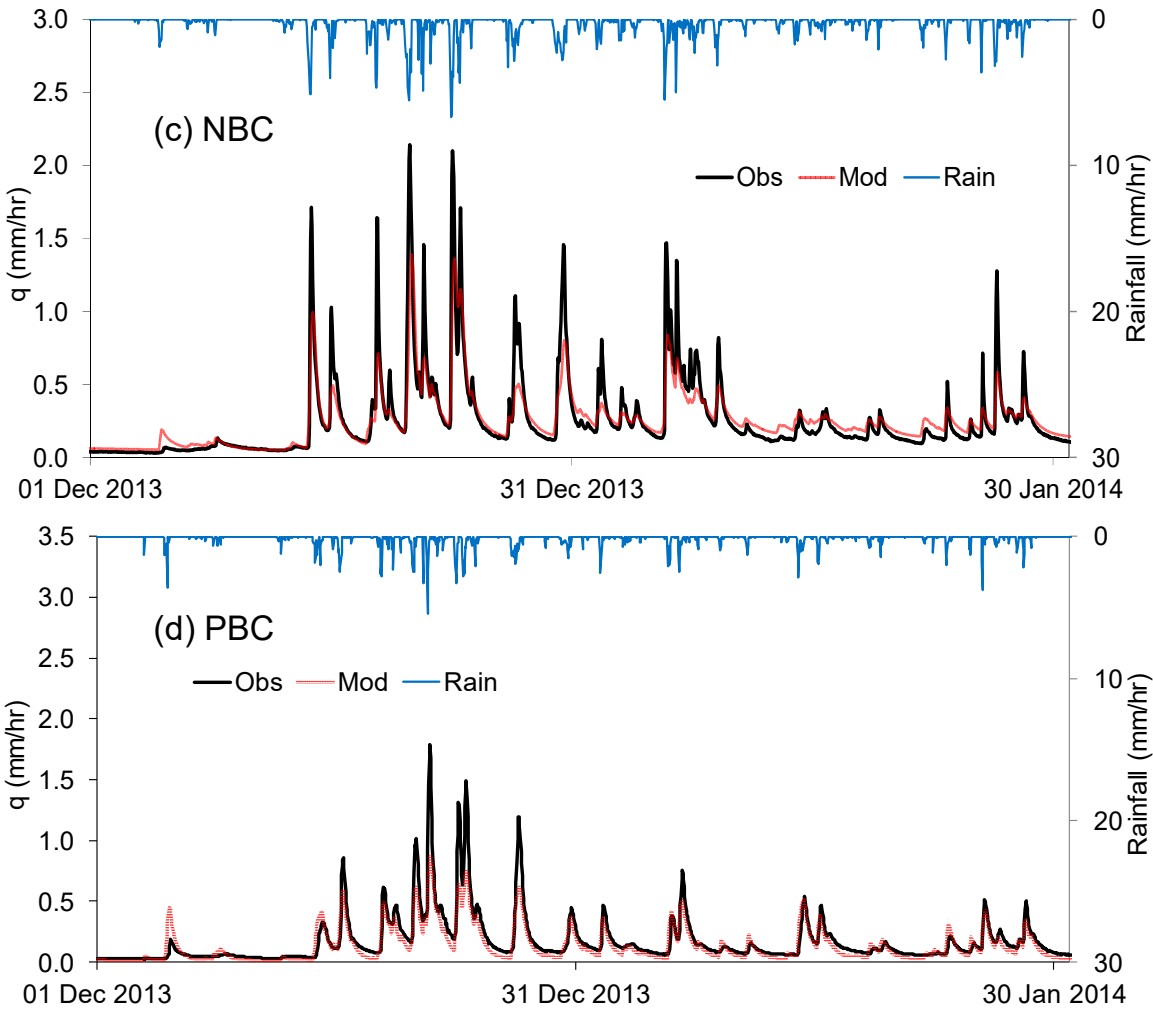

**Figure 6.** Time series plots of Observed and Modelled specific discharge (q) and rainfall at (**a**,**c**) NBC (**b**,**d**) PBC outlets: panes (**a**,**b**) showing (part of) calibration period (April–September 2012); panes (**c**,**d**) showing (part of) validation period (December 2013–January 2014).

**Table 3.** Shows a range of goodness of fit statistics for flow, TP and TRP concentrations for the whole study period, for each catchment. Positive bias indicates the model overpredicts, negative bias indicates the model underpredicts.

| Statistic | NBC (Period 1) | NBC (Period 2) | PBC (Period 1) | PBC (Period 2) |
|---|---|---|---|---|
| MBE(Bias) Q (%) | 0.36 | −0.16 | 6.5 | 17 |
| NSE Q (-) | 0.83 | 0.82 | 0.81 | 0.78 |
| LE (Bias) TP (%) | 11. | 46 | −5.0 | −8.8 |
| NSE TP (-) | 0.15 | 0.04 | 0.32 | 0.28 |
| LE (Bias) TRP (%) | −3.1 | 16 | −0.2 | −23 |
| NSE TRP (-) | 0.24 | −0.36 | 0.01 | −0.38 |

Modelling Events

The observed time lags and hysteresis between $Q_p$ and the maximum TP and TRP concentrations during events were observed to be quite complex and variable as discussed above (see Figure 5); for example, in the PBC both positive and negative lags were observed for TP and TRP. Due to the CRAFT's lumped structure, it cannot capture these complexities or simulate different patterns of hysteresis, however, through the attenuation store (in the surface runoff pathway) it is possible to delay Q together

with PP (currently assigning different lags to nutrients from Q is not possible) although the peaks of Q and PP will still be in-phase. The fast subsurface component of the CRAFT also provides the user with the ability to inherently specify a lag in the TRP concentrations—this is controlled by the $K_{ss}$ parameter in the model, a larger value of this parameter will introduce a shorter lag time and vice versa. The best that can be achieved is an effective mixed signal concentration form the subsurface soil (SS) component, i.e., an average of the old and new flow for the whole event. Future model structures could try to reproduce the E-D threshold, but this may be very event dependent. Here, we will summarise the significant differences between (near) surface runoff, surface stormflow and groundwater fluxes on the final TP concentrations.

### 3.3. Comparing Catchment Behaviour

#### 3.3.1. Using Event Forensics

In both the NBC and PBC, the flow pathways that transported TP to the catchment outlets were first thought to be dominated by surface runoff, as evidenced by the high runoff coefficients [26]. However, there were some major differences in the observed load and concentration dynamics in the two catchments (as identified above in Section 3.1 through the event forensics). In terms of the observed differences in P dynamics, the following deductions were made:

- TRP event loads in the PBC were nearly always higher (about 80% of the events analysed) than TUP loads during events. In the NBC the TUP loads were always higher than the TRP loads as the TP load was generally comprised of 60%–70% TUP. This indicates that there may be an additional source of TRP in the PBC close to watercourses that can become active during events including readily transported pools of SRP.
- Figure 3 shows that, from events in the NBC, TUP was the prevalent species of P being transported in preference to TRP. In the PBC this pattern was only observed during one event (17th September 2012) which had a much higher than average event maximum concentration of TUP. Thus, this event had the lowest maximum TRP concentration of all the events observed in the PBC for reasons that are not clear but may be due to seasonal factors (e.g., additional uptake by microorganisms during late summer and early autumn).
- It follows on (in the NBC) that "D Type" events may have been occurrences where the fast subsurface flow pathway was being damped by dilution (so TRP concentrations stayed close to their baseflow pre-event value) but "E-Type" events were ones where the fast subsurface flow pathway (with a higher TRP concentration than the slow groundwater flow pathway) was more active, probably due to having a larger soil water pool.
- Some clear dilution of TRP based on the temporal pattern of event TRP concentrations was observed during some "E-D Type" events in the NBC but not so in the PBC. This dilution was probably due to new (event) water predominating in shallower soils. We assume that this water contains a high proportion of rainfall with a much lower TRP concentration than older water that is in contact with nutrient enriched pools at or just below the ground surface.

#### 3.3.2. Using "Informed" Model Results

The above comparison now forces us to consider the differences and how they might be captured and communicated to end users. The subsequent sections will focus mainly on P loads and yields, and since other modelling studies [26,31,32] have only attempted to model P loads this seems reasonable. The model only prescribes three smart export coefficients (pathways), this means that the subsurface recharge and non-linear displacement/dilution dynamics are not simulated. It is therefore necessary to deduce how much of the observed catchment behaviour can still be captured.

The smart export coefficients for the NBC and PBC baseline model simulations can provide additional information on the likely pathways for P export that the event forensics alone or simple baseflow separation cannot achieve. The pairs of graphs shown above in Figure 7 are for the two

different time periods that were modelled; the pair (panes (a) and (c)) being from the calibration period results in 2012; the pair (panes (b) and (d)) are from the six-month validation period in 2013–2014. They represent a "snapshot" of parts of the entire record with wet conditions and with many runoff events; in the first period these mostly occurred throughout summer and early autumn, in the second period these occurred in winter.

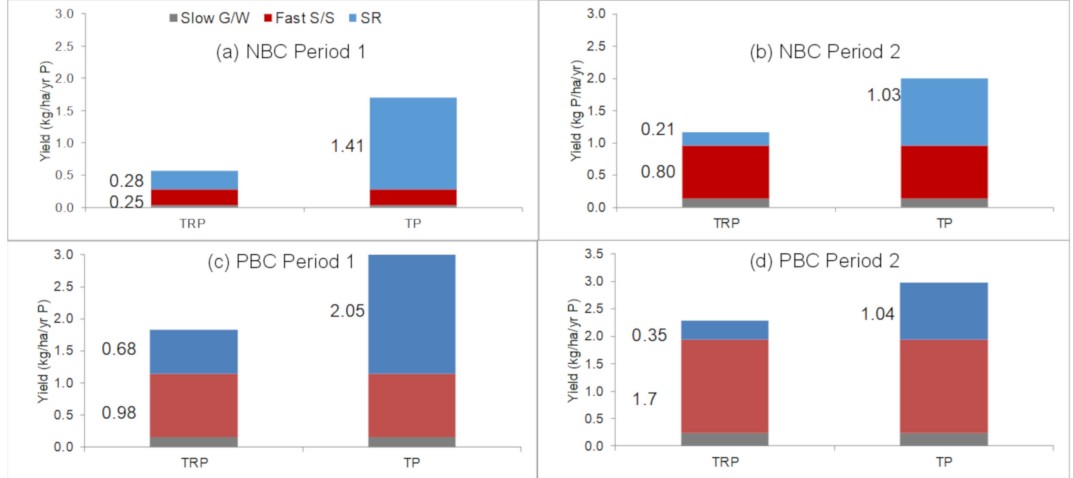

**Figure 7.** Smart export coefficients for entire time period from the NBC (**a,b**), PBC (**c,d**). Left panes (**a,c**) indicate values from model calibration (period 1), right panes (**b,d**) from validation (period 2). The smart export coefficients for the fast subsurface (SS) and surface runoff (SR) flow pathways are shown on the plots (export coefficients for Slow G/W were around $0.1$ kg P ha$^{-1}$ yr$^{-1}$ in all cases and are not shown).

The results clearly indicate the following phenomena. Firstly, the patterns of the modelled P yields are in broad agreement with the observed P yields in that the PBC was predicted to export a higher overall yield of TP than the NBC, which was mainly because the SRP yield was much higher from the fast subsurface store (soil water). These results were achieved by ensuring reasonably tight criteria when modelling in terms of insisting on a mass balance error (observed—modelled yield) of less than ±15% and a positive Nash and Sutcliffe efficiency (NSE) for both TP and TRP concentrations, for the calibration period, and desirable (see Table 3) for the validation period. Secondly, the flow pathways that the model determined differ between catchments, in that the PBC has a dominant fast subsurface ("Fast SS") pathway (contributing SRP) but the NBC was dominated by the (faster) surface runoff pathway ("SR") which was mostly contributing PP but also some TRP (e.g., particulate forms of reactive P that are sediment-bound).

Thirdly, the six-month validation period in 2013–2014 was wetter than the calibration period, and exported more of PP, TRP and TP from both catchments than the calibration period. The smart export coefficients for the NBC indicate that the fast subsurface export of TRP was four times that of the calibration period. The proportion of P exported as TRP by the fast subsurface pathway in the PBC was also twice that of the calibration period with a surprisingly low export by the surface runoff pathway during this period (note that these totals have been scaled up to represent annual totals, for comparison of periods of different length).

Figure 8 shows the event yields of TUP (i.e., PP) (exported by surface runoff) and TRP (exported by the fast subsurface and slow groundwater flow pathways) exported for selected events (where these were common to both sub-catchments). It can be seen, once again, that the fast subsurface pathway is of similar importance to the surface runoff pathway in the PBC but not so much in the NBC where the export of TUP by surface runoff dominated and dilution occurs more readily. The selected events do have quite different characteristics in that the smart export coefficients (shown in Figure 7) averaged over a much longer time period cannot be identified as easily. Event NP6 (24/9/2012) has already been

discussed above in some detail as one of the largest events observed during the entire period, and in both catchments the split, according to the model, of TUP from the surface runoff pathway and TRP from the fast subsurface pathway was broadly similar with the TUP yield being higher than the TRP yield. The implications are mainly for any policy that affects only the storm events only or the annual yields (such as nutrient loading rates).

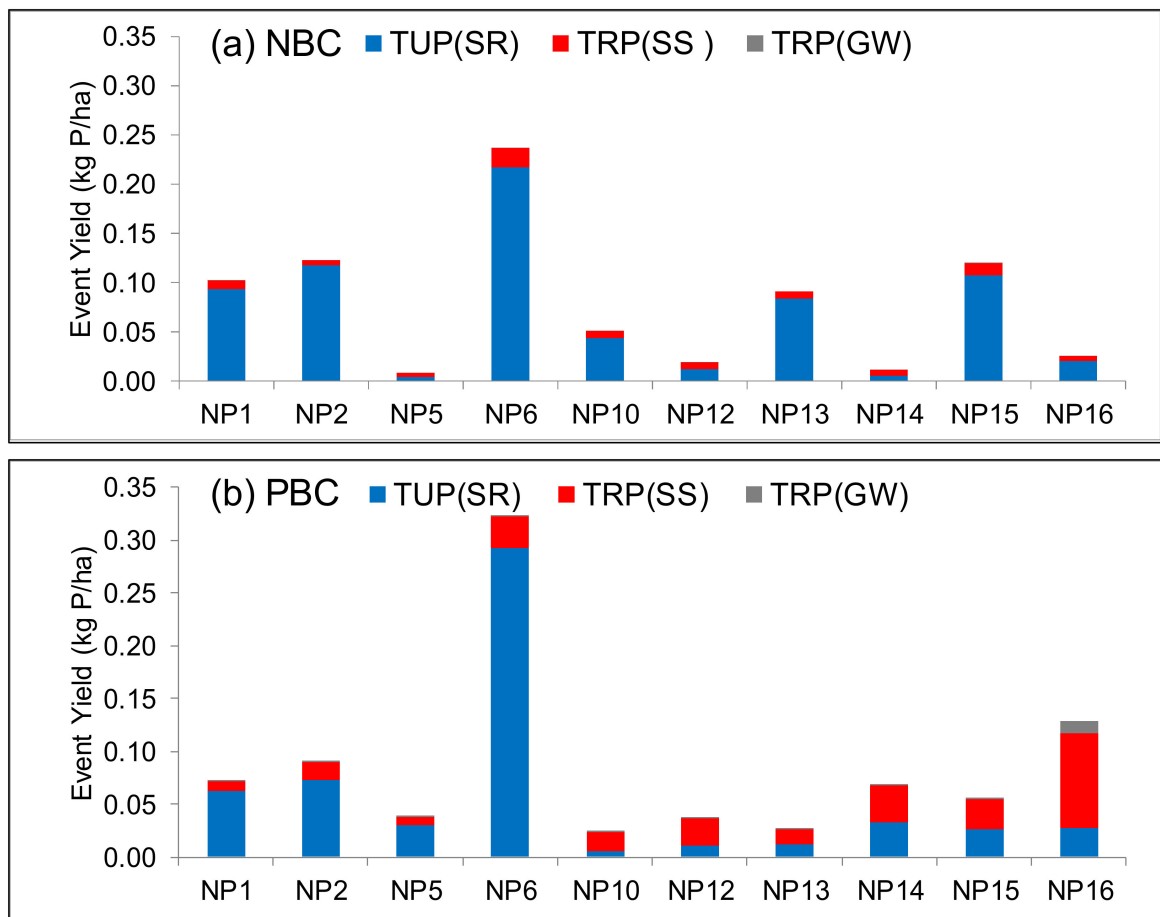

**Figure 8.** Event Yields for individual common events in (**a**) NBC, (**b**) PBC, events denoted by "NP<(Event number>" refer to text for details.

In order to examine the model findings in more detail, results from a single event (NP2 28/6/2012) are shown in Figure 9. The modelled P fluxes are shown along with the observed TP fluxes. In the NBC (pane (a)) the model predicted that only 14.5% of the modelled TRP was exported from the fast subsurface pathway. In the PBC (pane (b)) the model results indicate that a similar percentage of the total TRP was exported via this pathway (14%), the remaining TRP being exported via the surface runoff pathway in both sub-catchments (the slow groundwater contribution was negligible). A comparison between modelled and observed TP loads reveal that the model predicted the overall load reasonably accurately, but struggled to match the lag time observed in the PBC (in NP2 both the TP and TRP peak concentrations lagged $Q_p$ by +2 and +3 hours respectively). In the NBC, the model performed reasonably well in matching both the observed TP load and the timing, although in this case the model introduced a small lag that was not observed in the TP load time series in this catchment, indicating that in reality the TP load (the PP component of which) was probably transported in the rapidly responding surface runoff during the event rather than via field drains or interflow through the upper soil layers. The CRAFT cannot explicitly model the effect of field drains and adding additional flow pathways to the model is probably not justified due to the risk of introducing equifinality. Thus, lumping field drainage with surface runoff is assumed to hold true in these catchments.

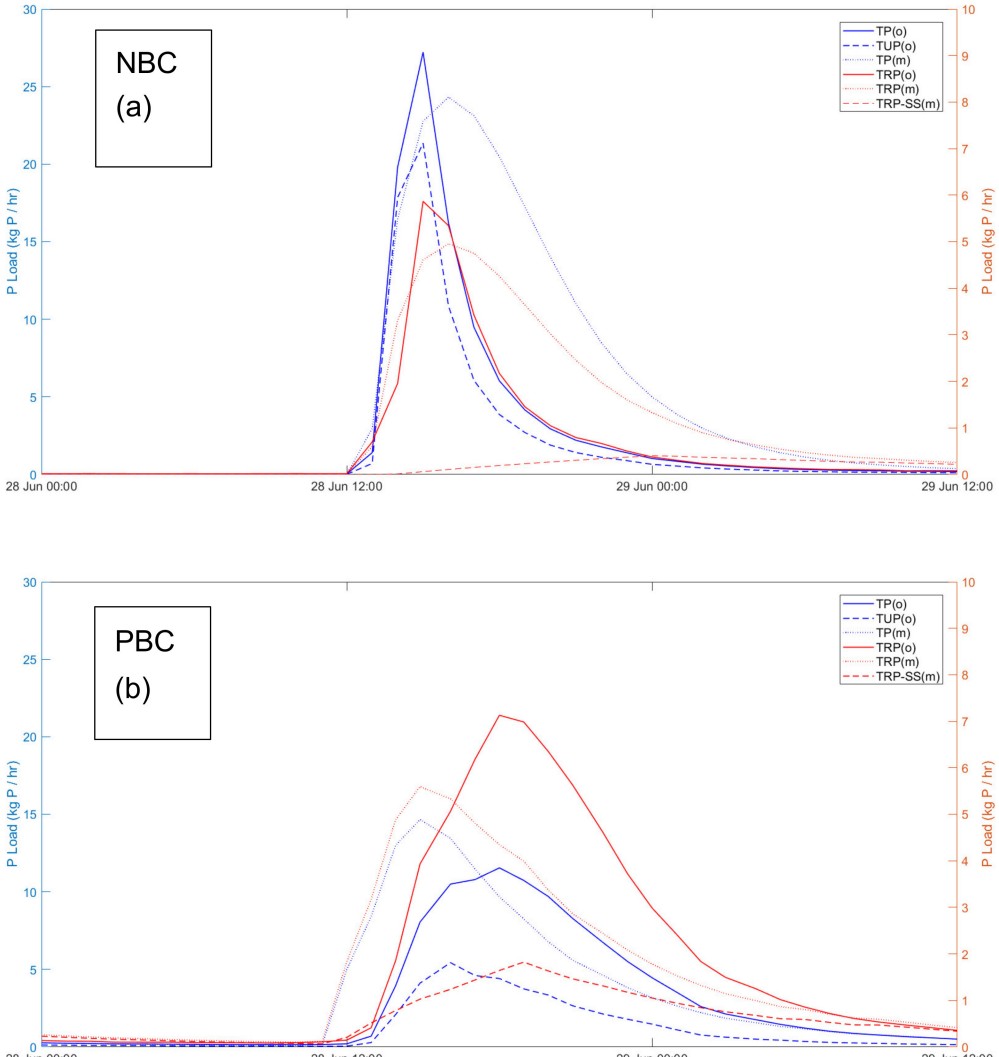

**Figure 9.** Plots showing modelled and observed P loads during event NP2 on 28/6/2012. Modelled TP, TUP and TRP loads are totals from all three flow pathways (denoted by "(m)"). Modelled TRP loads via the fast subsurface (SS) pathway are also shown by the dashed red line. Observed TP, TUP and TRP loads are denoted by "(o)". Pane (**a**) shows data from the NBC, pane (**b**) from the PBC. Note that the TRP load is plotted on the right-hand y axis, other loads on the left-hand y axis.

The contribution of the modelled TUP to the TP load during the NP2 event was quite different and much greater in the NBC than in the PBC as the plot of this flux time series depicts, this was in broad agreement with the event forensics, which for this event indicated that maximum TUP concentrations were 0.84 (NBC) vs. 0.4 (PBC) mg P L$^{-1}$. Observed TUP loads were estimated as 71.9 kg P (NBC) vs. 43.7 kg P (PBC) by subtracting the TRP load from the TP load, which is in agreement with the relative values of the observed concentrations and modelled loads in the two catchments. Moreover, if we assume that the SRP fraction of the TRP loads was 86% (NBC) and 94% (PBC), then the percentage of the observed TP loads exported as SRP during this event can be estimated. These percentages were 23.8% (NBC) and 58.3% (PBC), which shows a striking difference between the two catchments.

The EC values in the PBC groundwater samples were around 200 μS cm$^{-1}$ higher than the EC values recorded at the outlet prior to the "E and ED Type" runoff events (Figure 3). This may suggest that the baseflow observed at the PBC outlet was not entirely sourced from the same groundwater as the samples, which agrees with the findings of Allen's report [34] in that the deeper groundwater is not connected to the Pow Beck. This may also explain the very low deep groundwater contribution to

the total flow and P export. Further investigation of groundwater chemistry in the PBC is probably necessary to fingerprint sources of nutrients in order that measures can be taken to reduce concentrations of soluble P.

The accuracy of the modelled P concentration and loads during NP events for the other 16 storms requires more analysis. Figure 10 shows the accuracy of the model by showing a comparison of the observed and simulated TP and TRP loads and the goodness of fit criteria (NSE) for each storm. Table 3 suggested that the model is much weaker on P simulation compared to Q, but Figure 10 suggests that the dominant patterns are being emulated. The goodness of fit of each storm (N.B. no event calibration occurred) are overall high, but there are occasional storms that are non-behavioral and not well simulated. PBC (top right) has more scatter indicating more error than in NBC (top left).

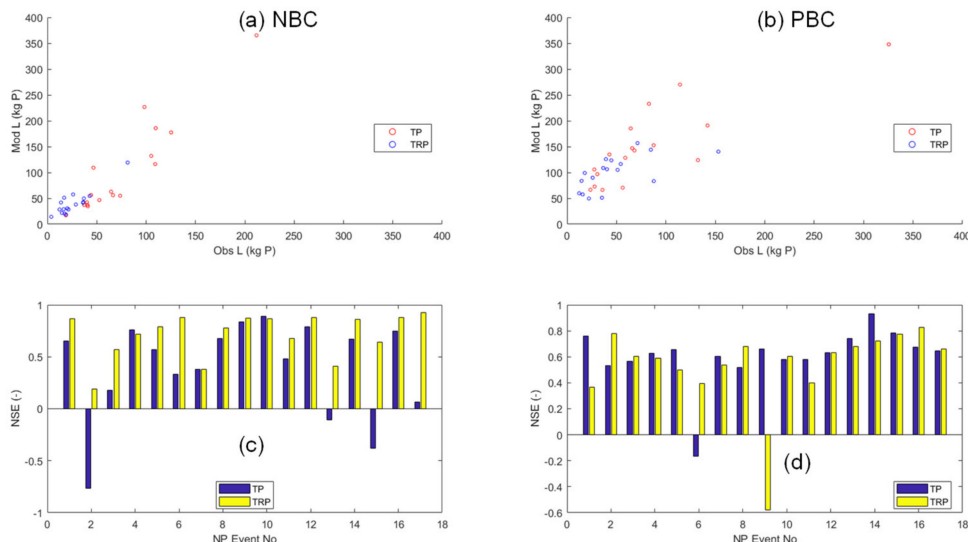

**Figure 10.** A comparison of (1) the observed and simulated TP and TRP loads for NBC (pane (**a**)) and PBC (pane (**b**)); (2) the NSE metric for each of the 17 storms analysed (NBC (pane (**c**)), PBC (pane (**d**))).

### 3.4. Summary

These findings are significant in terms of understanding P pathways and designing mitigation strategies, as the smart export coefficients varied considerably by pathway depending on the period being modelled, mainly due to variations in hydrological conditions. Seasonal variations could explain some of the differences, although when viewed as a whole, the time series of observed events in all categories contained a reasonably balanced proportion of events in all four seasons, with no one season being particularly wetter than another (in fact, the winter of the period 2011–2012 was very dry, as was the first half of 2013 prior to a series of events in late summer). Therefore, unravelling seasonality from the record would be difficult as cold and wet conditions leading to high P export can occur at any time of the year in the Eden.

In both catchments, the event forensics in conjunction with the findings of Ockenden et al. [26] suggested that the near surface runoff pathway (here including field drains and ditches) represents a major flow pathway for exporting particulate forms of P in the NBC, and to a lesser extent the PBC. The anti-clockwise hysteresis identified by Perks et al. [25], in events in the NBC for both TRP and TP, suggested that an event subsurface flow pathway, including lateral interflow through the upper soil layers, may dominate during events there. Ockenden et al. [26] suggested that the fast subsurface flow (including interflow through the subsoil) pathway is most important for mobilizing high concentrations and loads of TRP in both catchments due to the short lag time (two hours or less) observed between the peak Q and peak TRP concentration. In this study, these findings would concur with a subset of events with anticlockwise TRP hysteresis (Figure 5), e.g., NP2 (28th June 2012) in the PBC (Figure 3: third row, left). The role of enrichment on the rising limb for "E and ED Type" events was, in most

cases to elevate TP levels, this was followed by dilution that reduced the TRP concentrations (e.g., NBC. Figure 3: first row, left) but not the TP concentrations in the NBC due to a continued supply of PP. Damped signals from the "D-Type" events in the NBC, and to a lesser extent the PBC, showed the impact of old and new water mixing (21st December 2012 events, Figure 3: second and fourth rows, left). The use of EC data provides a potential low-cost solution for identifying event dynamics as the measurement technology is far cheaper than the bankside equipment required for continuous measurement of nutrients.

The CRAFT can capture most of these fluxes using three flow pathways, if near surface runoff and the drain flow are lumped with overland flow, and subsurface event flow includes a mixture of old and new water. "Plume" or "non-behavioural" events cannot be simulated. It was expected that the modelling study could help elucidate the flow pathways that were the most important during events in both catchments for both TP and TRP. Although the modelling results are not conclusive, it is clear that, overall, the near surface runoff pathway is the most important of the three for transporting P, especially for TP in the NBC (in the form of PP), but the fast subsurface pathway is more important for both catchments in terms of TRP, the bulk of which (around 90%) is exported in the form of SRP [25]. The smart export coefficients calculated by the model reflect this (Figure 7). The negative lag (clockwise hysteresis) observed between peak Q and both TP and TRP in the PBC (e.g., NP5) also indicated that near-stream or deeper soil water sources of P were being mobilised in the PBC. These findings reflect the need to: (i) target both the near surface runoff pathway and overland flow than just overland flow (surface runoff), as water is moving in and out of the top soil in the PBC and through field drains into ditches; (ii) target agricultural pools of soluble P located close to watercourses or deeper in the soil water column that originate from livestock farming operations over the inter-storm periods (such as feeding rings, manure piles, leaking slurry tanks). It may be necessary to develop an "expert" classification of farm types, dominant soil and groundwater characteristics (for example using Hydrology of Soil Types (HOST) [39] classes). This classification coupled with observed hydro-meteorological inputs may be enough to prioritise and target P loss via different flow pathways and thus influence policy makers.

## 4. Conclusions

A modelling study using the CRAFT with observations from two EdenDTC catchments showed that fast subsurface flow was the key pathway for P export (as SRP/TRP) in most events in the PBC, but the near surface runoff (flow) pathways dominated for PP export in both catchments, more so in the NBC. Overland flow and drain flow need careful interpretation as they are implicitly included in the fast "near surface runoff" flow pathway.

Many subtleties in the observed data are not be captured by the CRAFT, however, a good event-based catchment hydrological model accounts for most of the observed fluxes. Having an informed secondary "forensic" representation of the event type and the dominant flow pathways allows for much greater understanding of the likely fluxes in a catchment. Hence, we can inform the likely export coefficients for farmed landscapes based on several flow pathways, and we can infer the likely impact of new farming policies on changing these pathways. The model is robust at this spatial and temporal scale, and could form the basis of scaling up and addressing broader catchment management policies. The identification of flow pathway "smart" export coefficients is a key outcome of this approach, and it adds to similar export coefficient analyses that have been focused on annual losses per crop or farm type classifications.

Implications for management have already been identified, that showed, for example, that infiltrating near surface fluxes into the soil profile could reduce PP but will yield much more TRP at the catchment outlet [21]. Buffer zones and wetlands could process (biologically) SRP and trap some PP. Near surface fluxes and PP loss could be addressed through sediment traps and P recovery. These phenomena can be captured in CRAFT for a range of future management strategies, and moreover, for a wide range of storm magnitudes. Flow pathway management, particularly targeting storm events, must be considered on farmland to significantly reduce P loss rates.

While this study infers that forensic analysis of individual headwater catchments is expensive and time consuming, the knowledge gained is considerable and the fluxes observed can be captured in relatively simple and scale appropriate MIR models. These models can then form the basis of decision support tools that should prove useful for the policy makers. The observation strategy used here could be greatly influenced in the future by new technologies such as combined water level, turbidity and EC meters. Crucially, the move to a more event level observational capability is needed.

**Supplementary Materials:** The following are available online at http://www.mdpi.com/2073-4441/12/4/1081/s1, Figures showing all the common "NP<Event number>" events in the two catchments: Figure S1 (NBC) and Figure S2 (PBC).

**Author Contributions:** "Conceptualization, R.A. & P.Q.; Methodology, R.A. & P.Q.; Software, R.A.; Formal Analysis, R.A., N.B.; Investigation, R.A., P.Q., and N.B.; Data Curation, R.A. & N.B.; Writing-Original Draft Preparation, R.A. & P.Q.; Writing-Review & Editing. R.A., P.Q., N.B., and S.B.; Project Administration, S.B." All authors have read and agreed to the published version of the manuscript.

**Funding:** The Eden Demonstration Test Catchment (Eden DTC) research platform which supplied the monitoring data used in this paper is funded by the Department for Environment, Food and Rural Affairs (Defra) (project WQ0210) and is further supported by the Welsh Assembly Government and the Environment Agency.

**Acknowledgments:** Dr Ben Sturridge (University of Lancaster) supplied groundwater chemistry data for use in this study.

**Conflicts of Interest:** The authors declare no conflict of interest.

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
