# Peer review of "Identifying Flow Pathways for Phosphorus Transport Using Observed Event Forensics and the CRAFT (Catchment Runoff Attenuation Flux Tool)"

_water, doi:10.3390/w12041081_

Round 1
Reviewer 1 Report
Dear authors,
Thank you for your exploration of how we can better identify and understand the transport of phosphorus and its concentration. Here are some suggestions that should improve the quality of your paper.
ABSTRACT should be shortened. I suggest to remove text from the line 17 to the line 24 in Abstract.
In KEYWORDS; instead of 'runoff' could be 'surface runoff' or more specific
In CONCLUSIONS should be a comment of authors' suggestion how to develop a strategy of actions before elevation of phosphorus concentration would be observed. You could also state how intelligent technologies and governance strategies could participate. I assume your aspect is not considering profit in any way.
Author Response
see attached which includes note to editor and comments for both reviewers

Reviewer 2 Report
The work addresses pathways for P export in two study sites in Northwest England by means of a modelling framework in an event-based separation scheme. The approach is key for better understanding critical pathways and also the influence of the precipitation occurrence timing, among other factors. The methods are appropriately described and the results are shown in a clear and precise way, and they depict the different cases observed, where modelling provides insight on the key drivers and major processes governing the outputs. The graphs are very efficient to assess this. My only concern is related, however, to the presentation of calibration/validation results. Despite the fact that showing all the events would unnecessarily increase the length of the manuscript, I think that the results in Fig. 9 indicate that some metrics are further needed to quantify the goodness of fit during the calibration and validation periods, on the two scales of the study, global periods and event-to-event basis. Dispersion graphs could be added, and some statistics of the P yields on an event basis would improve the understanding of the problem. The metrics’ needs refer to both the flow and P results of the calibration and validation periods. Following this, I recommend major revision so that chance to see this new information is provided, but from the already included results the suggested inclusion should be quite straightforward to the Authors. Some minor comments: 1. Lines 310-314. As the Authors remark, these values are somehow high; is there any previous analysis in the study sites that also show this? Alternatively, some comment on the reasons/processes behind would complement the information. 2. Lines 338-339. Is a comma is missing behind “of these”? Otherwise some words seem missing. 3. Lines 348-357 are repeated.Author Response
see attached which includes note to editor and comments for both reviewers

Round 2
Reviewer 2 Report
The Authors gave fully addressed my comments in this new version, which now includes relevant information for the comprehensive analysis of results. I would like to thank them for their work and attention.
I find the current version ready for publication.